# High-frequency gas effusion through nanopores in suspended graphene

I. E. Rosłoń [1,2✉], R. J. Dolleman [1,3], H. Licona[1], M. Lee[1], M. Šiškins [1], H. Lebius [4], L. Madauß[5], M. Schleberger [5], F. Alijani [2], H. S. J. van der Zant [1] & P. G. Steeneken [1,2✉]

Porous, atomically thin graphene membranes have interesting properties for filtration and sieving applications. Here, graphene membranes are used to pump gases through nanopores using optothermal forces, enabling the study of gas flow through nanopores at frequencies above 100 kHz. At these frequencies, the motion of graphene is closely linked to the dynamic gas flow through the nanopore and can thus be used to study gas permeation at the nanoscale. By monitoring the time delay between the actuation force and the membrane mechanical motion, the permeation time-constants of various gases through pores with diameters from 10–400 nm are shown to be significantly different. Thus, a method is presented for differentiating gases based on their molecular mass and for studying gas flow mechanisms. The presented microscopic effusion-based gas sensing methodology provides a nanomechanical alternative for large-scale mass-spectrometry and optical spectrometry based gas characterisation methods.

[1] Kavli Institute of Nanoscience, Delft University of Technology, Lorentzweg 1, Delft 2628 CJ, The Netherlands. [2] Department of Precision and Microsystem Engineering, Faculty 3mE, Delft University of Technology, Mekelweg 2, Delft 2628 CD, The Netherlands. [3] Second Institute of Physics, RWTH Aachen University, Aachen 52074, Germany. [4] CIMAP/GANIL, CEA-CNRS-ENSICAEN-UCN, blvd Henri Becquerel, Caen F-14070, France. [5] Faculty of Physics and CENIDE, Universität Duisburg-Essen, Duisburg 47057, Germany. ✉email: i.e.roslon@tudelft.nl; p.g.steeneken@tudelft.nl

Although graphene in its pristine form is impermeable, its atomic thickness causes it to be very permeable when perforated[1–3]. This is an advantageous property that has recently been exploited for filtration and separation purposes[4–13]. For sub-nm pore sizes, it has been shown to result in molecular sieving[14–16] and osmotic pressure[17] across graphene membranes. Besides filtration and separation, selective permeability might also provide a route toward sensing applications. In contrast to chemical[18] and work-function based[19] gas sensing principles, the advantage of permeation based sensing is that it does not rely on chemical or adhesive bonds of the gas molecules, which can be irreversible or require thermal or optical methods to activate the desorption of the bound gas molecules[20].

Here, we demonstrate that graphene membranes can be used to pump gases through nanopores at high frequencies, and that the motion of the graphene can be used as a probe of the gas dynamics. When gas molecules flow through pores that are smaller than the gas mean free path length, but larger than their kinetic diameter, their permeation is in the effusive regime. According to Graham's law[21], the effusion time constant $\tau_{\text{eff}}$ of gas escaping from a cavity of volume $V$ is proportional to the square root of the gas molecular mass $M$ and can be written in terms of the total effusive area $A$, the temperature $T$ and the universal gas constant $R$:

$$\tau_{\text{eff}} = \frac{V}{A}\left(\frac{2\pi M}{RT}\right)^{1/2}. \tag{1}$$

By using graphene membranes to pump gases[22] through focused ion beam (FIB) milled nanopores[23], we realise an attoliter effusive flow through an orifice. The permeation rate is determined from the frequency ($\omega$) dependent response function $z_\omega/F_\omega$ which is used to determine the gas-specific time-delay $\tau_{gas}$ between the optothermal actuation force $F_\omega$ and the membrane displacement $z_\omega$. We show that the permeation time-constants can be engineered by altering the number of pores, their cumulative area and by adding a flow resistance in the form of a gas channel in series with the pore.

## Results

Figure 1a, b show a scanning electron microscope (SEM) top-view of a graphene microdrum with a nanopore. Dumbbell-shaped cavities are etched in a silicon substrate with a 285 nm $SiO_2$ layer using reactive ion etching and covered by a two layer stack of graphene, creating drums with a diameter of 5 µm that are connected by a channel of 0.6 µm wide and 5 µm long. The bilayer graphene layer covers the full area of the dumbbell shaped cavity and gas that is trapped in the volume underneath the graphene can escape through the milled perforations. The frequency response curves of the membranes are measured using a laser interferometry setup (see Fig. 1c and Methods).

**Operation principle**. We now discuss how the frequency dependent mechanical response of the graphene drum to modulated laser actuation can be used to characterise the gas permeation rate through the porous membranes. In vacuum, the graphene membrane is solely actuated by thermal expansion, as a consequence of the temperature variations induced by the modulated blue laser. This effect has been extensively studied by Dolleman et al.[24] to characterise the heat transport from membrane to substrate. The temperature at the centre of the membrane $T(t) = T_{\text{ext}} + \Delta T$, where $T_{\text{ext}}$ is the ambient temperature, can be approximately described by a first order heat equation, where the optothermal laser power $\mathcal{P}_{\text{AC}}e^{i\omega t}$ is absorbed by the graphene membrane and thermal transport towards the substrate is approximated by a single thermal time constant $\tau_{\text{th}} = R_{\text{th}}C_{\text{th}}$

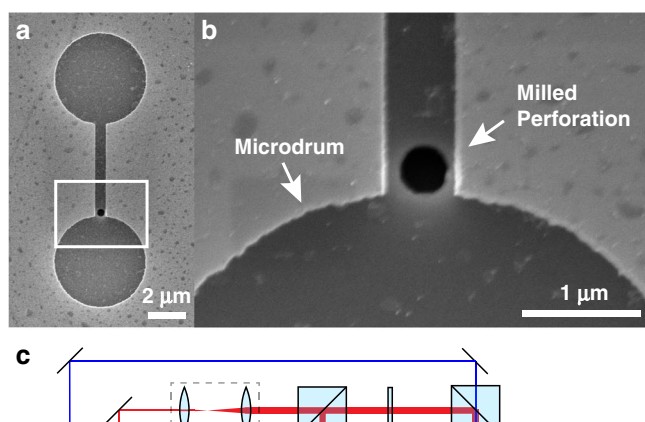

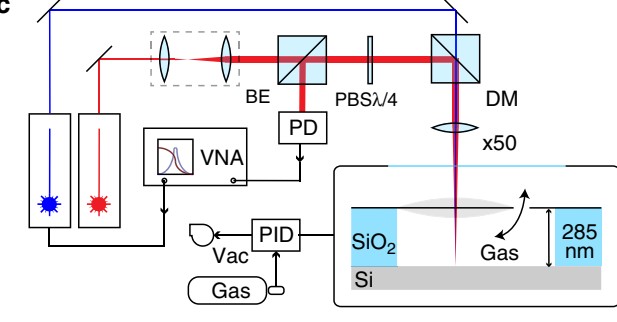

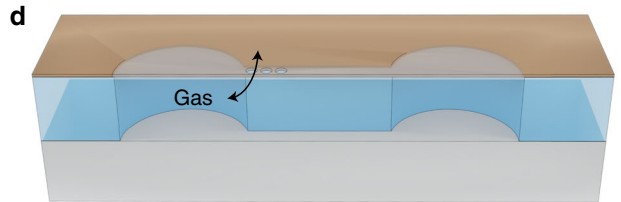

**Fig. 1 Suspended graphene device and measurement setup. a** Electron microscope image of a dumbbell shaped cavity covered by bilayer graphene. **b** A nanopore with a diameter of 400 nm is milled in the graphene by an ion beam in the channel that connects the two micro drums. **c** Interferometry setup used to actuate and detect the motion of the graphene micro drums. The red laser passes subsequently through the beam expander (BE), the polarised beam splitter (PBS) and the quarter-wave plate ($\lambda/4$), after which it is combined with the blue laser using a dichroic mirror (DM) and focused on a micro drum using a 50x objective. The readout is performed by a high-frequency photodiode (PD) that is connected to the Vector Network Analyser (VNA). The VNA modulates the power of the blue laser that actuates the membrane. Gas pressure inside the vacuum chamber is controlled by a PID controller. **d** Schematic of the device geometry and gas effusion path.

corresponding to the product of the membrane's thermal resistance and thermal capacitance:

$$\frac{d\Delta T}{dt} = -\frac{\Delta T}{\tau_{\text{th}}} + \frac{\mathcal{P}_{\text{AC}}}{C_{\text{th}}}e^{i\omega t}. \tag{2}$$

In the presence of gas, the pressure difference across the orifice $\Delta P = P - P_{\text{ext}}$ between the cavity pressure $P$ and the ambient pressure $P_{\text{ext}}$ can also be described by a differential equation. There are three contributions to the time derivative of the pressure $d\Delta P/dt$: gas permeation, motion of the membrane and laser heating of the gas in the cavity:

$$\frac{d\Delta P}{dt} = -\frac{\Delta P}{\tau_{\text{gas}}} + \gamma\frac{dz}{dt} + \frac{\mathcal{P}_{\text{AC}}}{C_{\text{gas}}}e^{i\omega t}. \tag{3}$$

Gas permeation out of the membrane with a time constant $\tau_{\text{gas}}$ gives a contribution $-\Delta P/\tau_{\text{gas}}$. Compression of the gas by the downward deflection $z$ of the membrane results in a term $\gamma dz/dt$, where $\gamma$ is a constant of proportionality. Heating of the gas due to power absorption of the modulated laser can be described by a

term $\frac{\mathcal{P}_{AC}}{C_{gas}}e^{i\omega t}$, where $C_{gas}$ is a constant relating thermal power to gas expansion.

A third differential equation is used to describe the mechanics of the membrane, which at low amplitudes experiences a force contribution proportional[24,25] to the pressure difference $F_P = \beta\Delta P$ and an effective thermal expansion force $F_T = \alpha\Delta T$:

$$m_{eff}\frac{d^2 z}{dt^2} + c\frac{dz}{dt} + kz = \alpha\Delta T + \beta\Delta P. \qquad (4)$$

Here, we describe the fundamental mode of motion at the centre of the membrane by a single degree of freedom forced harmonic oscillator with effective mass $m_{eff}$. The resulting system of three differential Eqs. (2)–(4) is solved analytically for frequencies significantly below the resonance frequency $\omega_{res} = \sqrt{k/m_{eff}}$, where terms proportional to $d^2z/dt^2$ and $dz/dt$ can be neglected, to obtain the complex frequency response $z_\omega/\mathcal{P}_{AC}$ of the membrane. A full derivation, solution and numerical simulation of the three differential equations can be found in the Supplementary Notes 1 and 2. The real and imaginary parts of the solution relate to the components of the displacement $z_\omega$ that are in-phase and out-of-phase with respect to the laser power modulation. The imaginary part of this expression is:

$$Im(z_\omega) = a\frac{\omega\tau_{th}}{1+\omega^2\tau_{th}^2} + b\frac{\omega\tau_{gas}}{1+\omega^2\tau_{gas}^2}. \qquad (5)$$

This equation is used to fit to the experimental data with $a$, $b$, $\tau_{th}$ and $\tau_{gas}$ as fit parameters. At frequencies close to the reciprocal permeation time $\omega_{gas} = 1/\tau_{gas}$ the imaginary part of the displacement displays a minimum, similar to the effect observed near $\omega_{th} = 1/\tau_{th}$ for the thermal actuation[24]. In the following, these extrema in the imaginary part of the frequency response will be used for characterising permeation and thermal time-constants.

**Response in gas**. A typical frequency response curve $z_\omega$ of a micro drum at a pressure $P = 250$ mbar in nitrogen gas is shown in Fig. 2a. The mechanical resonance occurs in the MHz domain, here at $f = 25.9$ MHz with $Q = 4.2$. Below the mechanical resonance, the imaginary response $Im(z_\omega)$ shows a characteristic dip-peak shaped curve for which the extrema are at 160 kHz and 2 MHz. These are assigned to the extrema of Eq. (5) corresponding to fit parameters $\tau_{th} = 1/(2\pi \cdot 2\,\text{MHz}) = 81$ ns and $\tau_{gas} = 1/(2\pi \cdot 160\,\text{kHz}) = 991$ ns.

To prove that one of the extrema is related to gas permeation, we study the dependence of the extrema on pressure. The measurements in high vacuum show only one extremum in the imaginary response, corresponding to a thermal time $\tau_{th} = 87$ ns, as shown in Fig. 2b. The disappearance of the dip at vacuum is a clear indicator that the dip is a result of gas interaction with the motion of the drum. Without a nanopore, the dip does not appear in any of the tested gases at any pressure (see Supplementary Fig. 6). This is evidence that the dip is actually a result of permeation through the nanopore. A small extremum is observed at $2 \cdot 10^5$ Hz in high vacuum such as in Fig. 2b, which is attributed to electrical cross-talk as discussed in[26]. This feature cannot be distinguished in Fig. 2a and is neglected in further analysis. The reference samples without perforations show only one thermal extremum with a similar time-constant $\tau_{th}$ to the perforated membranes. The second extremum, a dip in $Im(z_\omega)$, only appears for perforated membranes, and does not appear in high vacuum. Therefore, it is concluded that the dip in $Im(z_\omega)$ at 160 kHz in Fig. 2a is due to gas permeation with permeation time $\tau_{gas} = 1/(2\pi \cdot 160\,\text{kHz}) = 991$ ns.

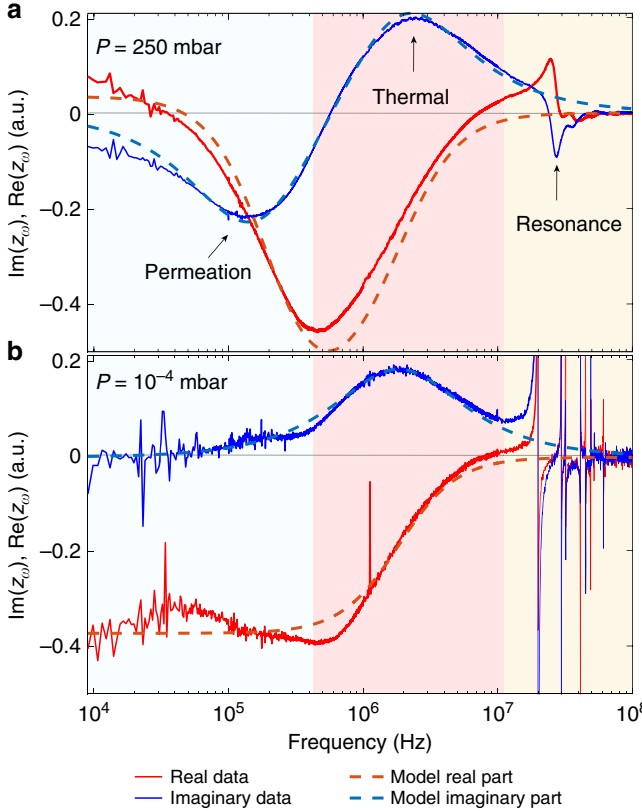

**Fig. 2 Graphene membrane motion $z_\omega$ (phasor) in gas and vacuum. a** Frequency response of the micro drum shown in Fig. 1 in nitrogen gas at $P = 250$ mbar showing the real (in phase, red curve) and imaginary (90 degree phase shift, blue curve) parts of the signal, $Im(z_\omega)$, $Re(z_\omega)$. Dashed curves indicate the fitted model response. **b** Measurement on the same micro drum at $P < 10^{-4}$ mbar shows that the permeation peak diminishes in vacuum and the maximum of the thermal peak shifts by 10% from 2 MHz to 1.8 MHz.

The permeation time constants $\tau_{gas}$ are extracted for a range of gases varying in molecular mass $M$ from 4 u (He) to 130 u ($SF_6$) in Fig. 3a. Figure 3b shows that the permeation time constant closely follows Graham's effusion law with $\tau_{gas} \propto \sqrt{M}$, as expected for gas transport through the nano pores. The slope of the linear effusion model is fitted to the data, and the grey area shows the 95% confidence interval of the fitted slope. This agreement demonstrates that the porous graphene membranes can be used to distinguish gases based on their molecular mass. A significant deviation between measurement and theory is only observed for He, which could be due to fitting inaccuracies related to the proximity of the thermal time-constant and mechanical resonance frequency peaks to the gas permeation related peak.

**Tuning of permeability**. The gas permeation time $\tau_{gas}$ can be tuned by varying the cumulative pore area, either by changing the number of pores or their size. This tuning can be useful, since too short time constants may lead to overlap between the $\tau_{gas}$ and $\tau_{th}$ peaks or even with the resonance peaks, whereas long permeation rates could be problematic in view of acquisition times.

Figure 4a demonstrates $\tau_{gas}$ tuning in drums with increasing number $n$ of 200 nm pores. The permeation time $\tau_{gas}$ is inversely proportional to the cumulative pore area $A$. The average reduction of $\tau_{gas}$ by a factor $2.26 \pm 0.5$ when doubling the number of pores from 1 to 2 is additional evidence that this time-constant is related to the permeation rate. The change in the

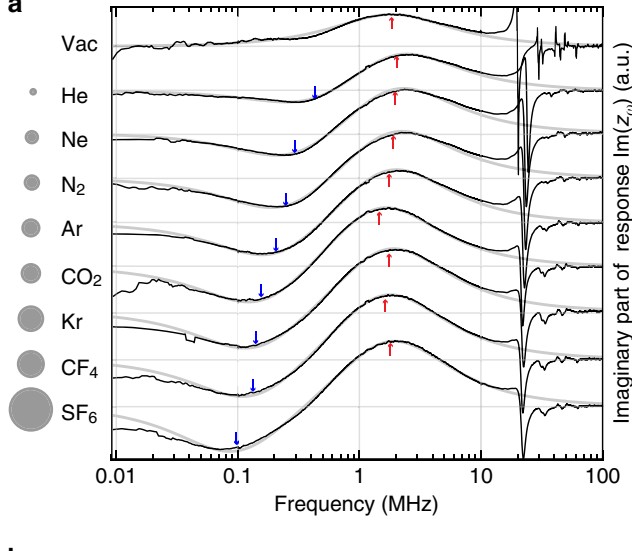

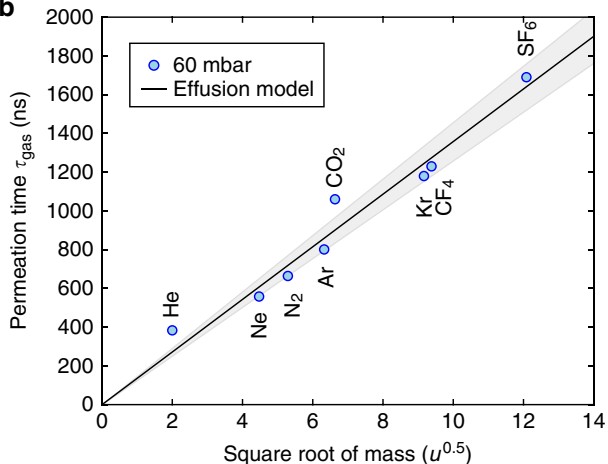

**Fig. 3 Dependence of $\tau_{gas}$ on gas molar mass. a** Measurements performed in high vacuum and in various gases at $P = 60$ mbar on a micro drum with a 400 nm pore. Eq. (5) (grey line) is fitted to the imaginary part of the measurement data (black line) and the red and blue arrows indicate the values obtained for $\tau_{th}$ and $\tau_{gas}$, respectively. The areas of the circles represent the relative mass of the gas particles. **b** The permeation time $\tau_{gas}$ increases linearly with the square root of the particle mass as predicted by Graham's law. The black line shows a fit of the measured values of $\tau_{gas}$ to Eq. (1) with $V/A = 2.71 \cdot 10^{-6} \pm 1.7 \cdot 10^{-7}$ m. The 95% confidence interval is shaded grey.

permeation time by a factor higher than two when doubling the number of pores might be caused by the fact that the two pores are located closer to the drum than the single pore, leading to a higher permeation rate. When increasing the number of pores to 3, the time-constant does not drop accordingly but by a factor $2.63 \pm 0.2$, indicating that other effects than pore effusion limit the permeation rate. The permeation area of 3 holes ($A = 9.4 \times 10^4$ nm$^2$) is 55% of the cross-section of the channel between the drums ($A = 17 \times 10^4$ nm$^2$). Therefore, the channel entrance acts as a significant additional obstacle for gas permeating through the pore, an effect that is further explored in the next paragraph. The pore placement in the channel and mechanical device-to-device variations could also be factors affecting the permeation time.

**Gas kinetics in a channel.** We investigate the gas kinetics further by placing the holes further away from the graphene drum, at the

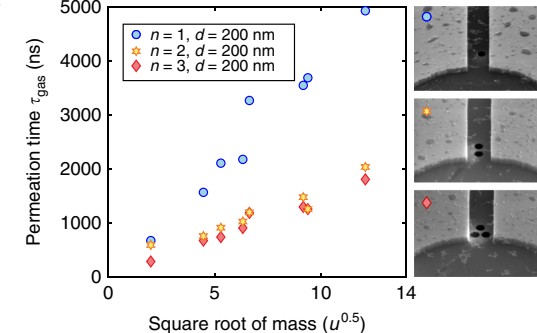

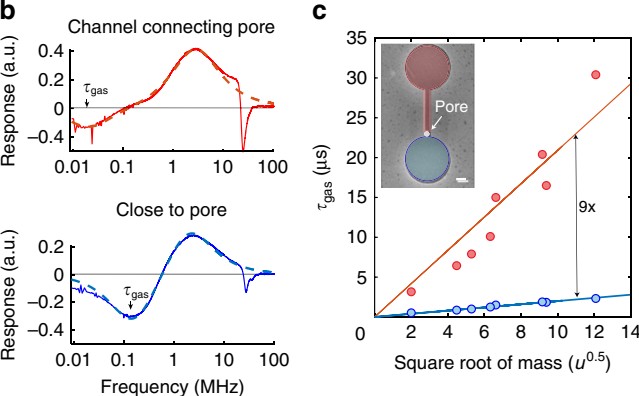

**Fig. 4 Tuning of the gas permeation time. a** Comparison of measurements between three different drums with increasing number of 200 nm pores. The permeation time $\tau_{gas}$ reduces with the cumulative pore area $A$, but saturates at 3 pores. **b** Measurements of Im($z_\omega$) with the laser aimed at the drum next to the pore and at the drum connected by a channel to the pore, respectively the blue and the red drum in the SEM inset in c, showing a large tuning of $\tau_{gas}$. **c** The gas permeation time of the drum close to the perforation is 9 times shorter than of the drum far away from the perforation. Inset: SEM image (false colour) of the two graphene drums connected by a channel with a 400 nm circular pore, scalebar = 1 μm. All measurements in this figure are performed at $P = 60$ mbar.

other end of the channel that connects both drums. The SEM inset of Fig. 4c shows a pore inside the channel, that is close to the blue drum, but far from the red drum. The rectangular, graphene-covered channel, with dimensions of $5 \times 0.6 \times 0.285$ μm$^3$, is in series with the pore for the red drum. It is found from Fig. 4c that the permeation time is 9 times longer for the red drum that is in series with the channel. The difference in permeation time is a measure of the transmission probability $\psi_r$ for gas atoms to pass through the rectangular channel. In the ballistic regime, the conductance and time-constant are given as the product of the time-constant of the aperture (the pore) and the transmission probability of the channel $\psi_r$ so that $\tau_{gas,close} = \psi_r \times \tau_{gas,far}$. The transmission probability through a rectangular channel can be calculated using the Smoluchowski formula[27] for which an useful approximation[28,29] is given by:

$$\psi_r = \frac{16}{3\pi^{3/2}} \frac{a}{l} \ln\left(4\frac{b}{a} + \frac{3}{4}\frac{a}{b}\right), \qquad (6)$$

where $a = 285$ nm and $b = 600$ nm are the cross-sectional dimensions and $l = 5$ μm is the length in the direction of gas flow. This formula predicts a 12% transmission probability for our geometry, in close agreement with the experimental value of 11% that is found from the ratio between the slopes of the blue and red solid lines in Fig. 3. We can conclude that ballistic transport is taking place in this nano channel.

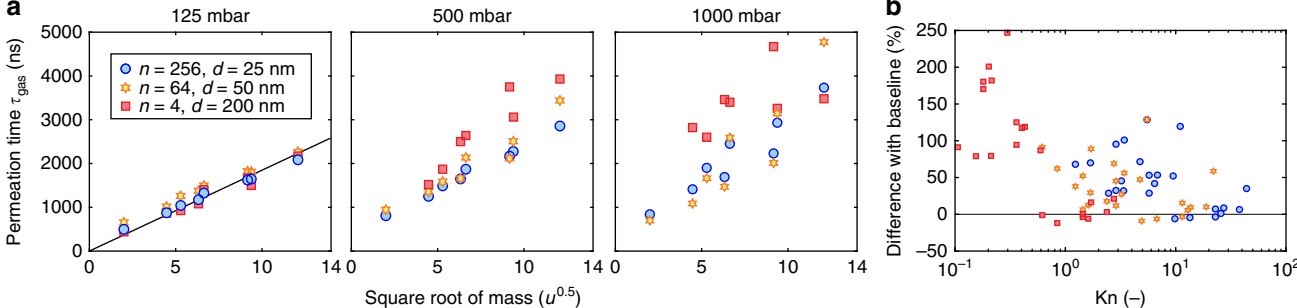

**Fig. 5 Transition from Knudsen to Sampson flow.** To compare different Knudsen numbers, the pore diameter $d$ and number of pores $n$ are varied, maintaining constant total pore area. Permeation times are shown in the transitional region between Knudsen and Sampson flow at a Kn range of 0.1–100. **a** At $P = 125$ mbar all devices follow Graham's law, as indicated by a fit to a straight line through the origin. At higher pressures, the measurements deviate from a straight line. **b** The difference with respect to the black baseline in **a** is calculated for all measurements shown in **a**. As Kn drops below 1, the permeation time increases and Graham's law does no longer describe the values of $\tau_{gas}$ correctly.

**Flow regimes**. The size of individual pores determines whether viscous Sampson or molecular Knudsen flow is taking place[30]. Figure 5 compares time-constants $\tau_{gas}$ in devices with equal cumulative area $A = 4\pi \cdot 10^4$ nm$^2$ and different pore diameters. At $P = 125$ mbar all devices show a linear relation between the square root of mass and the permeation time according to Graham's law. In contrast, at higher pressures where the mean free path length $\lambda$ becomes smaller than the pore diameter $d$ (Kn $= \lambda/d < 1$), in particular for the larger molecular masses and large pore sizes, the linear dependence disappears. In the transitional region between Knudsen and Sampson flow, classical effusion no longer correctly describes the flow and viscosity effects lead to larger values of $\tau_{gas}$ than predicted by Graham's law. This increase is in line with studies on pipe and channel flows[31], which show a maximum in the permeation time near Kn $= 1$ where the transition from Knudsen to Sampson flow occurs.

**Thermal transport**. Besides permeation, thermodynamic sensing can be achieved by observing changes in the thermal time constant in a fashion similar to Pirani gas sensors. In general the gas conducts heat better at higher pressures, and it does so also for molecules with a smaller molecular mass and higher molecular velocity. However, by analysing the values of $\tau_{th}$ that are determined from measurements like in Fig. 3a, it appears that the thermal conductivity of the gases is a less precise route toward gas sensing than the permeation based method shown in Fig. 3b. Further experimental results for the thermal time constant can be found in Supplementary Note 3.

**Discussion**

We have studied the effect of nanopores on the dynamics of graphene membranes. When gas is admitted to the nanodrums, it is found that a time delay appears between the membrane position and force (Fig. 2a), and that it does not only depend on the size and number of pores (Fig. 4a), but also on the type of gas (Fig. 3). This time delay is not observed in drums without nanopores (Supplementary Fig. S8). It is therefore attributed to permeation of gas through the nanopores and thus provides a method for studying nanoscale gas kinetics based on measuring the permeation time-constant $\tau_{gas}$ of gases through pores in bilayer graphene membranes. The method is based on high-frequency pumping of gases through nanopores. Due to the nanometre pore sizes, permeation is governed by effusion, such that permeation rates are inversely proportional to the square root of the molecular mass of the gas. By optothermal driving, the gas in the cavity below the graphene membrane is pressurised and pumped through the porous membrane. At angular driving frequencies close to the inverse of the permeation time

constant ($\omega = 1/\tau_{gas}$), a dip in the imaginary part of the frequency response appears which is used to characterise the gas species based on their effusion speed. By changing the number of pores and pore diameter using FIB, the time constants can be adjusted to a desired range. The presented measurement method is used to study gas flow through a microchannel at the transition from Knudsen to Sampson flow, where we observe an increase in the permeation time. This work shows that the extreme flexibility and permeability of suspended porous membranes of 2D materials can be used as an interesting platform for studying kinetics of gases at the nanoscale.

**Methods**

**Sample fabrication**. Dumbbell-shaped cavities are etched in a silicon substrate with a 285 nm SiO$_2$ layer using reactive ion etching, creating cavities with a diameter of 5 μm that are connected by a channel of 0.6 μm wide and 5 μm long. A stack of two chemical vapour deposited (CVD) monolayers of graphene is transferred over the cavity with a dry transfer method by Applied Nanolayers B.V. and subsequently annealed in an argon furnace. Nanoscale circular pores with diameters varying from 10 nm to 400 nm are milled through the suspended CVD graphene using a focused gallium beam FEI Helios G4 CX[10]. Pores are created in the channel instead of the drum, as directly milling on the drum reduced signal quality. Supplementary Note 4 discusses experiments on a circular single-layer graphene drum with perforations created directly on the drum. Supplementary Note 5 dicusses mechanical deformations introduced by milling of nanopores.

**Laser interferometry**. Two lasers are focused with a 1.5 μm spot size on the sample in a PID controlled pressure chamber. A red laser ($\lambda_{red} = 632.8$ nm) is used for detection of the amplitude and phase of the mechanical motion, where the position-dependent optical absorption of the graphene results in an intensity modulation of the reflected red laser light, that is detected by a photodiode[32]. A power-modulated blue laser ($\lambda_{blue} = 405$ nm), which is driven by a vector network analyser (VNA) at frequencies from 9 kHz to 100 MHz, optothermally actuates the membrane motion[33]. The incident red and blue laser powers are 2 mW and 0.3 mW, respectively. A calibration measurement, in which the blue laser is directly illuminating the photodiode, is used to eliminate systematic parasitic delays in the system[24].

**Data availability**

The data that support the findings of this study are available from the corresponding authors upon request.

**Code availability**

Code supporting this study is available at a dedicated Github repository: https://github.com/IrekRoslon/NatureCommunications.

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

## Acknowledgements

The authors thank Applied Nanolayers B.V. for the supply and dry transfer of bilayer graphene. This work is part of the research programme Integrated Graphene Pressure Sensors (IGPS) with Project Number 13307 which is financed by The Netherlands Organisation for Scientific Research (NWO). The research leading to these results also received funding from the European Union's Horizon 2020 research and innovation programme under Grant Agreement No. 785219 and 881603 Graphene Flagship and within the FLAG-ERA project NU-TEGRAM. F.A. and I.R. acknowledge financial support from the European Union's Horizon 2020 research and innovation programme under Grant Agreement 802093 (ERC starting grant ENIGMA). M.S. and L.M. acknowledge funding by the Deutsche Forschungsgemeinschaft (Project C5 within the SFB 1242 Non-Equilibrium Dynamics of Condensed Matter in the Time Domain (project No. 278162697) and SCHL 384/16-1 (project No. 279028710).

## Author contributions

I.E.R., R.J.D., H.Li., M.L., L.M., M.S., H.S.J.van der Z. and P.G.S. conceived the experiments. I.E.R. and H.Li. performed the laser interferometry experiments. H.Li., M.L. and M.Š. fabricated and inspected the FIB perforated samples. H.Le and L.M. prepared the ion bombarded samples. I.E.R., R.J.D. and H.Li analysed and modelled the experimental data. The project was supervised by F.A., H.S.J.van der Z. and P.G.S. The paper was jointly written by all authors with a main contribution from I.E.R. All authors discussed the results and commented on the paper.

## Competing interests

The authors declare no competing interests.
