## [Peer Review File · Nature Communications]

REVIEWER COMMENTS

Reviewer #1 (Remarks to the Author):

Roston et al. present an interesting and well written study on gas effusion through suspended graphene.

However, clear and direct evidence to unambiguously assigning the observed time delay between the actuation force and the membrane mechanical motion (the permeation time constant) to actual transport of gas through the nanopore is essential and is currently missing from the manuscript. Without this validation the entire manuscript could be just be an exercise in correlations, without a definitive causation assignment to independently and unambiguously shore up the results reported here.

Some additional concerns are:

1. The authors should show a systematic data for atleast 4 different pressure for atleast 3 gases to rule out the influence of damping from gas pressure on the drums.
2. The authors should explain why the single layer graphene showed such poor signal but the signal improves dramatically with 2 layers graphene? In this regard maybe even a change in the title to reflect bilayer graphene is perhaps appropriate.
3. Permeation rates should be affected by the amplitude of deflection of the drums, can the authors provide evidence for this?
4. Can the authors provide AFM images showing the cleanliness of the graphene to rule out adsorption of gases on the contaminants on graphene and any influence of contaminants on the observations.
5. Why does the permeation time constant more than double for 2 holes but remains almost unchanged from 2-3 holes? Can the authors show evidence to rule out effects from mechanical deformations that may be different in the drums after introduction of 1 vs 2 vs 3 holes?
6. Figure 2 – re-word legend “imaginary data” to perhaps something more meaningful e.g. imaginary part of measured data?
7. In Figure 2, the thermal peak is not unchanged but shifts significantly with frequency. The authors should explain this and remove the incorrect claim that it does not change.
8. Figure 2, there is a 2nd extrema visible at $\sim 2 \times 10^5$ Hz? Why is this ignored in the explanation? Assuming this extrema exists then the whole model and fit is inaccurate.
9. Although it is not surprising that the permeation time constant scales with mass in effusion. Can the authors provide evidence that this effect is not from damping from gas molecules as well as gas graphene interactions?

10. Can the authors show evidence for this statement “When increasing the number of pores to 3, the time-constant does not drop accordingly, indicating that other effects than pore effusion limit the permeation rate, such as the time taken by the gas to reach the pores at the edge of the drum.”

11. Referencing is inadequate and several seminal papers in the field have not been cited.

Reviewer #2 (Remarks to the Author):

Manuscript by Roston et al discusses novel and very interesting platform to study gaseous dynamics at the nanoscale. The proposed approach is very interesting and seems solid. The manuscript is written very well. I recommend accepting the manuscript if authors present some additional measurements.

Authors varied many experimental parameters, but I would suggest to also vary reservoir volumes and measurements temperature to further support their claims.

Reviewer #3 (Remarks to the Author):

This work presents the effusion results of several gases via drilled nanopores in graphene layers. There is a complex experimental set up with high precision controls to ascertain the gas permeation constants. A large number of experimental tests were carried out at set points of pressures with several gases, and different sizes of nanopores. Mathematical models were derived to explain the physics associated with the experimental procedures. The gas effusion results fit well with the square root of molecular mass of the tested gases, thus making a novelty case as an alternative method for gas testing.

Therefore, this work shows novelty and warrants publication in Nat Comm subject to a moderate review as follows:

1) The position of the graphene stack containing two layers is not clear. Please clarify if the two layers are covering the entrance of the cavity between the two micro drums. This clarification is necessary in the paper.

2) What is the mean free path length for the two-layer graphene stack used in this work?

3) How many times were the gas testing experiment replicated for each testing point to ensure the confidence level of 95%?

4) Was the temperature gradient in equation (1) affected by different gases? For instance, helium does not generally adsorb on surfaces whilst CO₂ has a stronger adsorption particularly on carbon derived surfaces at room temperature. This reviewer wonders if any heat variation due to adsorption/desorption was neglected as measurement times are given in nanoseconds.

5) Why is the fitting of the effusion model limited to lower pressures only? Apart from the reasons provided, at higher pressures the concentration of gas is higher. Hence, the latter could affect the heat gradient associated with gas adsorption/desorption, and lead to scattering of results.

6) The legend in Fig. 5 is not clear. For instance 256 x 25nm could mean the number of nanoholes (256) and the nanohole diameter (25 nm). This needs clarification in the paper.

7) Supplementary note 4 – bottom paragraph page 5: minor typo “nanpore” to be edited to “nanopore”.

Response to Reviewers

Reviewer #1

General: “Rostoń et al. present an interesting and well written study on gas effusion through suspended graphene. However, clear and direct evidence to unambiguously assigning the observed time delay between the actuation force and the membrane mechanical motion (the permeation time constant) to actual transport of gas through the nanopore is essential and is currently missing from the manuscript. Without this validation the entire manuscript could be just be an exercise in correlations, without a definitive causation assignment to independently and unambiguously shore up the results reported here.”

We thank the reviewer #1 for bringing to our attention that the evidence for assigning this time delay to actual transport of gas through the nanopore is missing. We will first discuss the evidence we have, and then discuss how we have improved the manuscript to improve the clarity on this important point.

We attribute the time delay to gas permeation for the following reasons:

- 1. In vacuum there is no time delay related extremum observed, whereas when admitting gas this extremum appears. This can be seen from the vacuum data in Fig. 2b, where the blue curve doesn't show a large negative extremum like in Fig. 2a.*
- 2. If the number of pores increases, the time delay decreases (Fig. 4a).*
- 3. If the type of gas is changed at the same pressure the time delay changes (Fig. 3).*
- 4. In the absence of nanopores, the time delay disappears (figure has been added as Supplemental Figure 6).*

In the paper we show that these observations are in line with physics based predictions for permeation through nanopores by gas effusion. It seems that they cannot be accounted for by any other mechanism and thus they present clear and unambiguous evidence that the time delay is due to gas permeation through the nanopore.

We agree with the reviewer that not all evidence was presented clearly and thanks to his comment we have been able to clarify this. Besides the new figure S6, we also added this sentence to the discussion section: “We have studied the effect of nanopores on the dynamics of graphene membranes. When gas is admitted to the nanodrums, it is found that a time delay appears between the membrane position and force (Fig. 2a), that does not only depend on the size and number of pores (Fig 4a), but also on the type of gas (Fig. 3). This time delay is not observed in drums without nanopores (Fig. S6). It is therefore attributed to permeation of gas through the nanopores and thus provides a method for studying ...”

Comment 1: “The authors should show a systematic data for at least 4 different pressure for at least 3 gases to rule out the influence of damping from gas pressure on the drums.”

As requested by the reviewer, we provide a systematic data set of 4 different pressures for 3 gases in Supplementary Figure 7.

Comment 2: “The authors should explain why the single layer graphene showed such poor signal but the signal improves dramatically with 2 layers graphene? In this regard maybe even a change in the title to reflect bilayer graphene is perhaps appropriate.”

The single layer graphene has been treated in a different way than the 2 layer graphene, namely by heavy ion bombardment in a particle accelerator instead of focussed ion beam milling. We believe that difference in treatment is the main reason for the poorer signal obtained in case of single layer graphene. Although the methodology works both with 1 and 2 layer graphene, we used 2 layer graphene in most of our experiments because of the higher membrane yield and durability. The method works for both single layer and bilayer graphene which is summarized in the title by the word graphene. Therefore, we prefer to keep the current title.

Comment 3: “Permeation rates should be affected by the amplitude of deflection of the drums, can the authors provide evidence for this?”

We agree with the Reviewer that permeation rates should be affected by the amplitude of deflection of the drums, since permeation rates Φ of an effusive flow follow:

$$\Phi = \Delta PA / (2\pi MRT)^{1/2},$$

where ΔP is the gas pressure difference, A is the area of the pore, M is the gas molar mass, R is the universal gas constant and T is the absolute temperature. However, with our method we obtain a permeation time constant, which for effusive flow is given by:

$$\tau = \frac{V}{A} \left(\frac{2\pi M}{RT} \right)^{1/2},$$

which describes the decay time, which is independent of amplitude. This is similar to the voltage decay on a capacitor C that is shunted by a resistor. The current depends on the voltage, but the RC decay time is independent of that voltage. To clarify this in the main text, we have included the equation for the permeation time constant of an effusive flow.

Comment 4: “Can the authors provide AFM images showing the cleanliness of the graphene to rule out adsorption of gases on the contaminants on graphene and any influence of contaminants on the observations.”

We thank the reviewer for this question. SEM images (Fig. 1b) confirm that indeed some polymer contamination from the transfer process is present on the surface of the graphene. AFM images, which have been added as Supplementary Fig. 5, show that these spots have a thickness of less than 15 nanometers. The contaminant is most likely PMMA, which is known to only absorb CO_2 among the gases we have used. Adsorption of gases by these contaminants on the graphene surface can indeed increase the mass of the membrane. However, no large change in resonance frequency is observed in between gases, which indicates that this effect is relatively small. Moreover, the permeation time-constant is measured at frequencies much below the resonance frequency, where the membrane mass does not play a big role, and the permeation time constant is independent on the mass or stiffness of the membrane. For the same reason we find little differences between the time constants measured by single and bilayer graphene.

Comment 5: “Why does the permeation time constant more than double for 2 holes but remains almost unchanged from 2-3 holes? Can the authors show evidence to rule out effects from mechanical deformations that may be different in the drums after introduction of 1 vs 2 vs 3 holes?”

This is a good question. In a crude approximation the holes can be seen as N parallel resistors, with resistance $R \propto 1/N$ such that the RC time scales as $1/N$. So 1 vs 2 should result in a reduction of tau by a factor 2 and going from 2 vs 3 should result in a reduction by a factor 3/2. However,

the flow resistance is not only determined by the pores, and therefore an additional series resistor is expected to be present that should reduce these factors.

Moreover, in practice the different relative positions of the pores in the channel and their distance to the drum can also play a role (see SEM in Fig. 4a). Furthermore, fabrication induced device-to-device variations can also affect the permeation time. The FIB procedure does indeed produce mechanical deformations as is shown in the new Supplemental Figure 5, that might account for variations in the time constant. We conclude that it is therefore not only the number of pores, but also these other factors that can affect the permeation time. We thank the reviewer for these remarks and have clarified this in the main text.

“... other effects than pore diffusion limit the permeation rate. The permeation area of 3 holes ($A = 9.4 \times 10^4 \text{ nm}^2$) is 55% of the cross-section of the channel between the drums ($A = 17 \times 10^4 \text{ nm}^2$). Therefore, the channel entrance acts as an significant additional obstacle for gas permeating through the pore, which is further explored in the next paragraph. The pore placement in the channel and mechanical device-to-device variations could also be factors affecting the permeation time.”

Comment 6: “Figure 2 – re-word legend “imaginary data” to perhaps something more meaningful e.g. imaginary part of measured data?”

We thank the Reviewer for noting this. We have changed the legend text to “imaginary part of measured data”.

Comment 7: “In Figure 2, the thermal peak is not unchanged but shifts significantly with frequency. The authors should explain this and remove the incorrect claim that it does not change.”

We have changed the evaluating phrasing for the maximum of the thermal peak from “remains almost unchanged” into “shifts by 10% from 2.0 MHz to 1.8 MHz ”. We have added markings to indicate that the x-axis is logarithmic. We point out that the permeation peak disappears in high vacuum, whereas the thermal peak is still clearly present, although shifted in frequency.

Comment 8: “Figure 2, there is a 2nd extrema visible at $\sim 2 \times 10^5 \text{ Hz}$? Why is this ignored in the explanation? Assuming this extrema exists then the whole model and fit is inaccurate.”

The reviewer is right in sharply observing that in Fig. 2b there is a small peak in data measured in vacuum at $\sim 2 \times 10^5 \text{ Hz}$. Since there is no gas present in this measurement in vacuum, this peak cannot be related to gas permeation. This feature is rather an effect of electrical crosstalk as discussed by Dolleman et al. in Supplemental Figure 4 of [1]. There, it is shown that increasing the power of the blue driving laser reduces the amplitude of the small feature in the data measured in vacuum at $\sim 2 \times 10^5 \text{ Hz}$. Moreover, connecting the ground of the RF cables to the metal of the optical table changes the frequency and amplitude of this feature. Therefore, this feature is designated to electrical cross-talk rather than motion of the graphene membrane.

We add a notice for the reader: “A small extremum is observed at $2 \times 10^5 \text{ Hz}$ in Fig. 2b, which is attributed to electrical cross-talk as discussed in [1]. This feature cannot be distinguished in Fig. 2a and is neglected in further analysis”.

[1] Dolleman, Robin J., et al. "Transient thermal characterization of suspended monolayer MoS₂." *Physical Review Materials* 2.11 (2018): 114008.

Comment 9: “Although it is not surprising that the permeation time constant scales with mass in effusion. Can the authors provide evidence that this effect is not from damping from gas molecules as well as gas graphene interactions?”

We have performed measurements on graphene membranes without pores. In those membranes we did not observe a permeation time constant proving that the permeation time-constant is related to the nanopore, instead of gas damping or gas-graphene interactions (see new Fig. S6).

Comment 10: “Can the authors show evidence for this statement “When increasing the number of pores to 3, the time-constant does not drop accordingly, indicating that other effects than pore effusion the permeation rate, such as the time taken by the gas to reach the pores at the edge of the drum.”

This comment is related to comment 5. If the time-constant does not drop with increasing number of pores, then in our reasoning that logically implies, that its value is determined by other effects than the number of pores. We provide the hypothesis that the time for the gas to reach the nanopores contributes to limiting the total time-constant, for which we provide evidence in Fig. 4b, by varying the length of the channel to reach the pore.

Comment 11: “Referencing is inadequate and several seminal papers in the field have not been cited.”

We have included all papers that we have read and that were relevant for this work. If some papers have missed our reading attention, we sincerely apologise. For completeness, we have included the following recent review articles which came out just this year: “Experimental and theoretical exploration of gas permeation mechanism through 2D graphene (not graphene oxides) membranes” and “Applications of nano-porous graphene materials – critical review on performance and challenges”. However, if the reviewer has specific other suggestions for important papers in this field we are certainly interested to add them as references.

Reviewer #2

General: “Manuscript by Rostoń *et al.* discusses novel and very interesting platform to study gaseous dynamics at the nanoscale. The proposed approach is very interesting and seems solid. The manuscript is written very well. I recommend accepting the manuscript if authors present some additional measurements. Authors varied many experimental parameters, but I would suggest to also vary reservoir volumes and measurements temperature to further support their claims.”

We thank the reviewer for his/her kind words. We agree with the reviewer that the method is well suited to vary more parameters than those included in our study: pressure, gas molar mass (M), pore area (A) and pore diameter. With our method we find a permeation time constant, which for effusive flow is given by:

$$\tau = \frac{V}{A} \left(\frac{2\pi M}{RT} \right)^{1/2},$$

Studying the only 2 parameters left, temperature (T) and volume (V) would indeed be a logical choice for further research, and can provide further support for the method. Although we will consider this suggestions for future research, we will not include such measurements in this manuscript, because the time required would result in a large delay in publication of this manuscript and we do not expect these measurements to change the main conclusions. Additional measurements to strengthen our claims are provided in the form of a systematic

data set of 4 different pressures for 3 gases in in Supplementary Figure 7 and data on an unperforated membrane in Supplementary Figure 6 .

Reviewer #3

General: "This work presents the effusion results of several gases via drilled nanopores in graphene layers. There is a complex experimental set up with high precision controls to ascertain the gas permeation constants. A large number of experimental tests were carried out at set points of pressures with several gases, and different sizes of nanopores. Mathematical models were derived to explain the physics associated with the experimental procedures. The gas effusion results fit well with the square root of molecular mass of the tested gases, thus making a novelty case as an alternative method for gas testing. Therefore, this work shows novelty and warrants publication in Nat Comm subject to a moderate review"

We thank the reviewer for his/her kind words.

Comment 1: "The position of the graphene stack containing two layers is not clear. Please clarify if the two layers are covering the entrance of the cavity between the two micro drums. This clarification is necessary in the paper."

We thank the reviewer for pointing out this un-clarity. The bilayer graphene layer is covering the full area of the dumbbell shaped cavity: the 2 drums and the channel in-between them. The only area without bilayer graphene is the nanopore. We have made this more clear in the main text and included a 3D render of the geometry in Fig. 1d to clarify the exact placement of the graphene and nano pores. We have added the sentence: "The bilayer graphene layer is covering the full area of the dumbbell shaped cavity and gas that is trapped in the volume underneath the graphene can escape through the milled perforations."

Comment 2: "What is the mean free path length for the two-layer graphene stack used in this work?"

The two graphene layers are touching each other, so there is practically no mean free path in between them. The improved geometry that we have added in Fig. 1d will help to clarify this.

Comment 3: "How many times were the gas testing experiment replicated for each testing point to ensure the confidence level of 95%?"

In figure 3b in the main text the permeation time for 8 different gases is plotted versus molar mass. The slope of these data points is fitted by a straight line through the origin. The 95% confidence level in the slope is determined and indicated by the grey area in Fig 3b. Therefore, the shown confidence interval is based on 8 data points. We have clarified this in the text: "...the grey area shows the 95% confidence interval of the fitted slope."

Comment 4 "Was the temperature gradient in equation (1) affected by different gases? For instance, helium does not generally adsorb on surfaces whilst CO₂ has a stronger adsorption particularly on carbon derived surfaces at room temperature. This reviewer wonders if any heat variation due to adsorption/desorption was neglected as measurement times are given in nanoseconds."

We did study the effect of gases on the temperature difference ΔT between membrane and substrate, and this data is shown in Fig. S3 and S4c. Both the type of gas and the pressure are seen to influence ΔT via gas cooling of the membrane. It might be that changes in the thermal conductivity of the graphene by gas adsorption also affect ΔT , but we cannot resolve/prove such effects.

Comment 5: “Why is the fitting of the effusion model limited to lower pressures only? Apart from the reasons provided, at higher pressures the concentration of gas is higher. Hence, the latter could affect the heat gradient associated with gas adsorption/desorption, and lead to scattering of results.”

We thank the reviewer for his/her constructive comment. Whether the effusion model is applicable depends on the Knudsen number $Kn = \lambda / L$. It relates the mean free path of gas particles (λ) to the dimensions of the pore (L). For the effusion model to apply, Kn should be larger than one. This is the case when pores are sufficiently small and/or pressure is sufficiently low. At larger pressures and for gases with larger molecular mass (smaller λ), $Kn < 1$ and the effusion model breaks down.

To address the question of the reviewer we provide additional data and fits in new Figure S7 at pressures up to 1000 mbar. It can be seen that the model presented in the paper does not fit the data as well at these pressures, which might be due to the fact that the system is not in the effusive regime anymore. Also the adsorption/desorption effects mentioned by the reviewer might complicate the situation here.

We thank the reviewer for the interesting question and will consider a more in-depth study of graphene gas permeation at higher pressures as a future research topic, for which more measurements and analysis will be required.

Comment 6 and 7: “The legend in Fig. 5 is not clear. For instance 256 x 25nm could mean the number of nanoholes (256) and the nanohole diameter (25 nm). This needs clarification in the paper. Supplementary note 4 – bottom paragraph page 5: minor typo “nanpore” to be edited to “nanopore”.”

We thank the reviewer for his/her careful reading and have changed the manuscript as suggested. The legend now reads ‘ $n=256$, $d = 25\text{nm}$ ’, for example.

REVIEWERS' COMMENTS

Reviewer #1 (Remarks to the Author):

The reviewer appreciates the response from the authors. The edits/clarification introduced to the manuscript and supporting information are adequate. Congratulations on this fine piece of scientific work.

Reviewer #2 (Remarks to the Author):

The authors satisfactorily addressed my concerns. I suggest publishing the manuscript.

Reviewer #3 (Remarks to the Author):

The authors have addressed all my questions adequately. It is recommended for this manuscript to be approved for publication in Nature Communications.